# Clinical Manifestations of Kawasaki Disease at Different Age Spectrum: A Ten-Year Study

**DOI:** 10.3390/medicina56040145

**Published:** 2020-03-25

**Authors:** Cristina Medeiros R. de Magalhães, Fernanda Coutinho de Almeida, Lenora Gandolfi, Riccardo Pratesi, Natália Ribeiro de M. Alves, Nicole Selleski, Renata Puppin Zandonadi, Eduardo Yoshio Nakano, Claudia B. Pratesi

**Affiliations:** 1Pediatric Rheumatology Unit, Children’s Hospital José Alencar, Brasilia 74083-330, Brazil; veredao@gmail.com; 2School of Medicine, Brasilia University Center—UNICEUB, Brasilia 70790-075, Brazil; 3Interdisciplinary Laboratory of Biosciences, School of Medicine, University of Brasilia, Brasilia 70910-900, Brazil; fcoutinhodealmeida@gmail.com (F.C.d.A.); lenoragandolfi1@gmail.com (L.G.); pratesiunb@gmail.com (R.P.); metodologiacientificaunb@gmail.com (N.R.d.M.A.); selleskinicole@gmail.com (N.S.); 4Post-graduate Program in Medical Sciences, School of Medicine, University of Brasilia, Brasilia 70910-900, Brazil; 5Post-graduate Program in Health Sciences, School of Health Sciences, University of Brasilia, Brasilia 70910-900, Brazil; 6Department of Nutrition, Faculty of Health Sciences, University of Brasilia (UnB), Campus Darcy Ribeiro, Asa Norte, Brasilia 70910-900, Brazil; renatapz@yahoo.com.br; 7Department of Statistics, University of Brasilia, Brasilia 70910-900, Brazil; eynakano@gmail.com

**Keywords:** Kawasaki disease, self-limited vasculitis, coronary artery lesions, global health issue

## Abstract

*Background and objectives:* The present study is the first known in Latin America to enroll a substantial number of Kawasaki disease (KD) patients with an extended follow-up. This study aimed to: (1) to expose the difficulties and delays in the diagnosis of KD in a developing country, (2) to describe and correlate the clinical features of this disorder with the children’s age at the time of disease onset, (3) to correlate the frequent lack of early diagnosis with a delayed application of appropriate treatment, and (4) to describe the outcome and eventual recurrences of KD in our region. *Materials and Methods:* Three hundred and one participants (183 males and 118 females) included in the study were diagnosed and, subsequently, clinically followed for ten years (January 2007 to December 2016) at the Pediatric Rheumatology Walk-in Clinic of the Children’s Hospital of Brasilia. *Results*: Episodes ranged from four months to two years. This rate of recurrence was well-above that disclosed by previous reports. Delay in diagnosis, in all age groups, caused an undesirable delay between the disease onset, the final diagnosis, and the administration of intravenous immunoglobulin (IVIG). KD recurred in 25 (8.3%) of the children during the first three years of follow-up. In seven patients, KD recurred twice, with an interval between episodes ranging from four months to two years. *Conclusions*: This rate of recurrence was well-above that disclosed by previous reports. In Latin America, aside from a handful of physicians and researchers, KD is being ignored. There is a pressing need to educate primary health care physicians and bring awareness to the fact that KD is not an exotic condition that affects only the Asian populations but a disorder that already exists among us and that frequently results in severe consequences.

## 1. Introduction

Kawasaki disease (KD) is an acute, febrile, self-limited vasculitis initially described in 1967 in Japan by Tomisaku Kawasaki [1]. It is the leading cause of acquired heart disease in children in developed countries and, therefore, increasingly has increasingly recognized worldwide as a matter of public health [2]. Lack of data is partially due to the lack of an adequate diagnostic marker [3]. The cause of KD is still undefined, and the prevailing hypothesis is that an immunologic reaction is brought about by one or more still-unknown triggers in genetically susceptible individuals. KD is characterized by a fever for five or more days and at least four of the five following clinical criteria: edema, erythema, and cracking of the lips, strawberry tongue, and/or erythema of oral and pharyngeal mucosa; bilateral bulbar conjunctival injection without exudate; rash: maculopapular, diffuse erythroderma, or erythema multiforme-like; erythema and edema of the hands and feet in acute-phase and/or periungual desquamation in the subacute phase; and cervical lymphadenopathy (≥1.5 cm in diameter), usually unilateral [4,5]. 

KD affects all racial and ethnic groups, and 80% of cases occur between the ages of six months and four years [6,7]. However, KD has been diagnosed in infants one month or less and also in young adults in their early twenties [5,8]. Diagnosis and treatment are frequently delayed in cases detected at an extreme age range. This may occur either because of incomplete manifestation, where patients may present atypical forms of the disease, failing to display all clinical diagnostic criteria or due to a reluctance by the attending physician to diagnose the disease outside the typical age range [9]. Children affected before or after the period in which KD is considered more prevalent are at higher risk for coronary artery abnormalities [7]. While higher risks may be partially due to delayed diagnosis and treatment, they can also be attributed to the immunologic immaturity of young infants. That may explain the high rate of aneurysms across all ethnic groups and poor outcomes seen even in infants with timely diagnosis and treatment [10]. In Brazil, a delayed diagnosis of KD by the primary physician is frequent even in age groups where KD is more prevalent. As found in a previous study from our laboratory, the most frequent misdiagnoses of KD were allergic reactions and exanthematous infectious diseases of infancy [11]. 

As previously noted, epidemiological studies focusing on KD are rare both in Brazil and throughout Latin America [3,12]. Data regarding the incidence of KD in Latin America is insufficient, and that is also true for data on morbidity and mortality, partially due to the lack of an adequate diagnostic marker [3]. In 2013, a Latin American Kawasaki Disease Network (Red de Enfermedad de Kawasaki en América Latina—REKAMLATINA) was established. REKAMALATINA initiates its historical path towards understanding the epidemiology of KD in children from Mexico, Guatemala, Honduras, El Salvador, Nicaragua, Costa Rica, Panama, Cuba, Dominican Republic, Puerto Rico, Colombia, Venezuela, Ecuador, Peru, Bolivia, Brazil, Paraguay, Uruguay, Chile, and Argentina. Today, REKAMLATINA is the largest multinational network studying the epidemiology of KD in children [12]. Nevertheless, as previously noted, epidemiological studies focusing on KD are still rare both in Brazil and throughout Latin America [3,11,12,13].

Therefore, the present study is the first known in Latin America to enroll a substantial number of patients with an extended follow-up. The objectives of the present study were: (1) to expose the difficulties and delays in the diagnosis of KD in a developing country, (2) to describe and correlate the clinical features of this disorder with the children’s age at the time of disease onset, (3) to correlate the frequent lack of early diagnosis with a delayed application of appropriate treatment, and (4) to describe the outcome and eventual recurrences of KD in our region.

## 2. Materials and Methods

The Ethics in Research Committee Foundation for Teaching and Research in Health Sciences at the University of Brasilia (Brasilia, DF, Brazil) approved the study in 2006 (protocol #1.037.234). Signed informed consent was obtained from parents or legal guardians, after an extensive explanation of the research purpose, at the time of KD diagnosis. 

All children included in the study were diagnosed and, subsequently, clinically followed during a one to ten-year-period, from January 2007 to December 2016, at the Pediatric Rheumatology Walk-in Clinic of the Children’s Hospital of Brasilia. The Children’s Hospital of Brasilia is a tertiary hospital subordinate to the Federal District Health Secretariat that is part of the Brazilian Unified Health System. The hospital mainly serves the low-income population from a vast network of secondary hospitals and primary health centers covering all districts and nearby regions. 

The diagnosis of KD, either in its complete or incomplete form, was made in accordance with the guidelines of the American Heart Association [4,7]. The diagnosis of atypical or incomplete forms da KD (i.e., fever for five or more days accompanied by less than four of the classic signs) relied on an abnormal C- reactive protein (CRP > 3.0 mg/dL) and erythrocyte sedimentation rate (ESR > 40 mm/h) in any infant or child with echocardiographic findings or the presence of certain laboratory features that may raise the clinical suspicion of KD (anemia for age, increased platelet count after day seven for fever ≥ 450,000/mm³, albumin ≤ 3.0 g/dL, elevated ALT level, WBC count > 15,000/mm³, and urine > 10 WBC/HPF) or the presence of coronary artery lesions (CALs) detected by echocardiography The diagnosis of atypical or incomplete forms was confirmed by two independent pediatric rheumatologists and by the exclusion of other possible diseases.

### 2.1. Subjects

The 301 participants (183 males and 118 females) that were followed during the 10 years were stratified into three groups according to the age at diagnosis. A small number of patients were lost to follow-up in the first year of the study and, therefore, they were not included in the study.

Group 1 was composed of 23 children (7.6%) age <1 year, Group 2 constituted by 177 children (58.8%) aged >1 year and ≤4 years, and Group 3 with 101 children (66.6%) aged >4 years.

All demographic and clinical data were recorded on a patient’s standard protocol that covered diagnostic criteria, response to therapy, unexpected complications during the disease course, laboratory results, echocardiographic findings, and other relevant information. After the diagnosis, all children were followed for at least one year and to up to ten years. 

According to the protocol, patients underwent standardized echocardiograms during the acute phase, after four to eight weeks during the convalescent phase, again after three and six months, and at the end of the first year from the beginning of the disease. After that, control echocardiograms were performed yearly. After the first year, CALs were defined by the criteria established by the Japanese Society of Pediatric Cardiology and Cardiac Surgery [14] and the American Heart Association scientific statement [7] which was the primary diagnostic reference when the study started. Aneurysms were classified as small (<5 mm internal diameter), medium (5–8 mm internal diameter), or giant (>8 mm internal diameter). Coronary arteries were classified as abnormal if the internal lumen diameter was >3 mm in children <5 years old or >4 mm in children ≥5 years old, if the internal diameter of a segment measures ≥1.5 times that of an adjacent segment, or if the coronary lumen is irregular. This classification is based on the Japanese Ministry of Health criteria [15] that were still adopted at the time of the beginning and during most of the study. All patients during the follow-up period were evaluated at least once for sensorineural hearing loss by evoked potentials. During outpatient follow-up, all patients underwent Doppler ultrasound of the entire abdomen to assess the abdominal arteries, and those who presented changes were submitted to abdominal angiotomography.

### 2.2. Data Analysis

The results were presented in frequency and percentages, and the three age groups were compared using the Pearson chi-square test with Monte Carlo approximation. The analyses were performed by the SPSS (Statistical Package for Social Science), and the tests considered a level of significance of 5%.

## 3. Results

Table 1 shows the time elapsed between the onset of symptoms, KD diagnosis, the administration of intravenous immunoglobulin (IVIG). And the number of children with incomplete or atypical KD. There were 1.55 males for each female patient. In all age groups (mean age 2.9 years), an undesirable delay occurred between the disease onset; the final diagnosis (group 1: 5–19 days, group 2: 5–25 days, and group 3: 5–30 days); and the administration of IVIG (group 1: 6–20 days, group 2: 5–26 days, and group 3: 5–32 days). An average of 10 days elapsed between the onset of the KD symptoms and IVIG application among the 301 participants (Table 1). Incomplete or atypical forms of KD were more frequent among infants less than one year and older than eight years.

The distribution of the main diagnostic criteria according to the age of disease onset is shown in Figure 1. Cervical adenopathy was the least observed disease sign, being present in only 53 (17.6%) of the 301 children and only one infant under the age of one year. The diagnostic criteria did not differ statistically between the ages of the disease (*p* > 0.05), except for cervical adenopathy (*p* = 0.004), indicating a higher frequency in the older age groups.

The presence of other abnormalities during the acute phase of the disease and other clinical findings during the acute and convalescent phases of the disease are shown in Table 2. An increased prevalence of CALs was observed among infants under the age of one year (12/23, 52.7%) and in children older than eight years (5/12, 41.6%). Among the 84 children that presented with CALs, the abnormalities were observed during the acute phase in 18 (21.4%), during the subacute phase in 49 (58.3%), and during the late phase of the disease in 17 (20.3%). Twenty-eight (33.3%) out of the 84 children affected by CALs disclosed lesions in more than one coronary artery. Few disclosed abnormalities in other arteries besides the coronary arteries, such as bilateral dilatations of both iliac arteries (three children), mesenteric artery (one child), aorta (two children), and brachial artery (one child). Aneurysm affected 42 (51%) of the 84 children. Giant aneurysm was rarely observed (13/84, 15.5%). Echocardiogram was performed either a few hours before or a few hours after the use of IVIG in 18 patients who had coronary changes in the acute phase. After the use of IVIG, the acute phase ended with the disappearance of the fever and other symptoms. The most frequent abnormality observed among all affected children during the acute phase was extreme irritability (233/301, 77.4%). In several children, to a greater or lesser degree, the irritability persisted during the following months, well into the late stage of the disease.

Behavioral abnormalities during the subacute and late stage of the disease were mainly characterized by irritability and aggressiveness that gradually disappeared from six months to one year after the acute phase; although, in rare cases, the parents reported a lasting learning deficit and alteration in their child’s behavior. Behavior abnormalities persist for six to twelve months. All children underwent brainstem-evoked response audiometry (BERA) six months after the disease onset. Hearing loss was observed in 41 (13.6%) children. The hearing loss was generally mild, bilateral (≤40 dB), and less frequent in children younger than one year. Abnormalities of the nervous system (seizures, chorea, facial paralysis, and learning deficit) were rarely observed. 

One child disclosed chorea during the subacute phase. Neuroimaging study revealed a hypodense lesion suggestive of ischemia near the caudate nucleus. Chorea was still present seven years after the disease onset, although partially controlled with valproic acid. Facial paralysis was observed during the acute phase in a child with complete regression during the subacute phase.

KD recurrence occurred in 25 children (8.3%) during the first three years of follow-up. In seven patients, KD recurred twice, with an interval between episodes ranging from four months to two years. In four of these patients, IVIG (2 g/kg) was applied after the tenth day of illness. Two other patients did not respond to the application of IVIG and required a course of corticosteroids. Small aneurysms appeared in three patients after the first recurrence and in two others after the second recurrence. 

Of the 301 patients in the study, 31 (10.2%) did not receive IVIG (2 g/kg). Of the 270 who received IVIG, 31 patients (11.4%) did not respond to the first infusion and only responded to treatment with the second IVIG infusion. Nine patients did not respond to both IVIG pulses (2 g/kg) and only became afebrile after methylprednisolone pulses (30mg/kg) for three days, and three patients also did not respond to methylprednisolone pulses and only became asymptomatic after using two doses of biological anti-TNF alpha (etanercept 0.8 mg/kg/week subcutaneously). No fatalities were recorded among the 301 children during the follow-up period. 

## 4. Discussion

KD presents a challenge to physicians working in Latin American countries, since it is an uncommon rheumatologic affliction that requires highly specific intervention to prevent serious or fatal sequelae but has a nonspecific clinical presentation in a critical period during which proper treatment should be instituted [16]. The age group in which the disease is most prevalent, between one and four years of age, coincides with the age at which childhood febrile exanthematic diseases are more common, which often hampers and delays the definitive diagnosis. Coronary aneurysms appear in 15% to 25% of cases [7], often silent, and may be recognized only years later in the event of sudden death or myocardial infarction [4,7]. There are many infectious diseases more prevalent than KD that present as acute febrile illnesses of childhood, confounding the diagnosis of KD [11,16].

Different from other studies, we conducted a study in which the children were followed during the entire period. Our data corroborated previous studies [4,7,16] showing the highest prevalence of KD among males (1.55 male:1 female) for all of the age groups (Table 1). Behavioral changes were observed with great frequency, not only in the acute phase but also in the chronic phase of the disease, which corroborates with findings published in other studies [13,17].

Previous studies were based on coronary arteriography of patients with a precedent history of CALs or an analysis of specimens from autopsies, cardiac transplants, or excised coronary aneurysms [18]. However, a more significant part of patients with KD do not exhibit CALs, and information regarding the functional condition of their coronary or other systemic arteries in the chronic stage of the disease are scarce. Histological changes and the functional condition of the coronary arteries in the chronic stage of KD were observed in patients [19,20,21]. Increased wall stiffness was detected by using the measure of a brachial-ankle pulse wave velocity years after the acute stage of KD in patients without apparent CALs during the acute phase of the disease [20]. Furthermore, a study assessed carotid and aortic intima-media thickness and pulse wave velocity, carotid distensibility, and diameter compliance in patients with and without a previous history of CALs [22]. This study showed that patients with KD had increased aortic intima-media thickness and reduced carotid distensibility notably (but not exclusively) in those with CALs more than two years after acute KD. Consequently, although KD has a self-limited acute phase, it must be considered an evolving process, with potential long-term deleterious consequences, requiring a prolonged follow-up.

It is challenging to ascertain how many children will present long-lasting structural sequelae of the kind described by Suzuki et al. [19] and Orenstein et al. [18] after the onset of KD, since many patients are undiagnosed or are lost to follow-up. Each of our patients was followed with periodic echocardiograms during a period between one and ten years. Seventeen (5.6%) disclosed a late onset of coronary artery dilatations during the chronic stage of the disease. Subtler abnormalities of the coronary arteries may not have been detected, since, in most cases, CALs were evaluated in consonance with the established criteria of the 2004 American Heart Association scientific statement [7], which were the current guidelines at the starting point of our study. Consequently, Z-scores were not routinely assessed in most of the affected children. Of the 301 patients, only 24 (8%) were of Asian descent (first and second generation). A study based on ethnic groups was not performed, since in Brazil, there is a large miscegenation of ethnic groups, making difficult the classification.

The rate of disease recurrence in this study (*n* = 25, 8.3%) was well above that disclosed by previous reports. Studies performed in the United States and Japan disclosed rates of recurrence between 2% and 4%. In our study, recurrences occurred mainly in the one to four-year age group. Seven patients had more than one recurrence with a time interval between the first and second episodes varying between three months and two years. We do not have an explanation for this increased rate of recurrence aside from the fact that this was a group of patients closely monitored by the same team during a relatively extended period. Few studies have evaluated neurological complications of Kawasaki disease; most of them are case reports [4,23,24]. These studies describe neurological changes based on an area of cerebral ischemia or involvement of the facial nerve with facial paralysis, cochlear nerve leading to ataxia, and impairment of the acoustic nerve with sensorineural hearing loss [4,23,24]. 

Several children (*n* = 41, 13.6%) disclosed neurosensorial hearing loss (NSHL). Our finding is similar to results obtained by other authors [23,24]. In a recent review [23], it was determined that some degree of NSHL affects approximately 36% of all patients with KD, and at least 14% of those children disclosed a persistent NSHL. In our cases, the hearing loss was generally moderate, of the order of 40 dB, and mostly bilateral, and in seven (17.1%), NSHL was still present after more than two years. It is remarkable that a single study, in one pediatric rheumatologic center in Brasilia/Federal District/Brazil. Brasilia is the capital of Brazil, located in the Federal District in the Midwest Region of Brazil. Brasilia is surrounded by low-income communities (favelas) that for the most part have experienced disordered growth and substandard housing conditions. The research was carried out at the children’s hospital, which is a public hospital located in Brasilia that is part of the Unified Health System and serves low and middle-income patients in Brasilia and the surrounding area.), disclosed a mean incidence of approximately 28 cases of KD per year. Although the Rheumatologic Unit of the Brasilia Children’s Hospital is considered a reference center for Kawasaki disease, several other hospitals in the city also admit and treat patients with this disorder. Consequently, if we consider the cases of KD admitted into other centers and cases not detected due to physicians’ lack of awareness of this disease and its frequent misdiagnosis for other exanthematic disorders, it is possible to conclude that the number of new cases of KD in our region is probably much higher. Our results show that our patients suffered from an unacceptable delay between the onset of KD symptoms and the administration of IVIG. As previously mentioned, the early application of IVIG is crucial for the abatement of the immune response and the prevention of significant coronary artery abnormalities. 

Community health centers in the Brazilian Unified Health system administer patients’ primary care. When strictly necessary, a patient will be referred to a network of secondary hospitals and, eventually, tertiary hospitals. This system, despite the excellence of its proposal and positive qualities, requires proper training and constant updating from the primary physicians to be able to discern patients at risk and promptly and adequately refer these patients for further diagnostic workup and treatment. We assume that the delay in the administration of appropriate treatment was firstly due to constant delays in diagnosis and, secondly, to the lapse between the child’s attendance by the health unit and the referral to a specialized center. Excluding the few cases seen primarily in our unit, patients referred by health centers and secondary hospitals reached our hospital on average 12 days after the onset of the first symptoms of the disease. 

## 5. Conclusions

Since KD is a vasculitis with a self-limiting acute phase in which a fever and clinical symptoms subsided even without specific treatment, it is worrisome to consider how many cases may have been misdiagnosed and progressed undetected, leading to future cardiac complications. 

There is no surveillance system for this disorder in Brazil; consequently, there is no data on its incidence, even in cases that are appropriately diagnosed. As in other diseases that were considered rare in the past and are now global health issues, in Latin America and other developing countries, detected cases of KD are probably just the tip of an unknown iceberg.

Globally, KD is recognized as a public health issue. However, in Latin America, aside from a handful of physicians and researchers, KD is largely ignored. There is a pressing need to educate primary health care physicians and bring awareness to the fact that KD is not an exotic condition that affects only the Asian populations but a disorder that already exists among us and that frequently results in severe consequences.

## Figures and Tables

**Figure 1 medicina-56-00145-f001:**
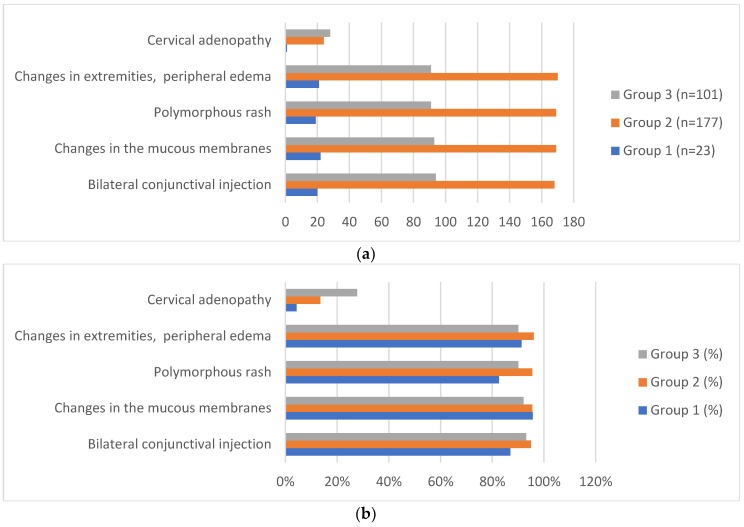
Distribution (**a**) and percentage (**b**) of the Kawasaki disease main diagnostic criteria in 301 children according to age at diagnosis. * Bilateral conjunctival injection, *p* = 0.332. Changes in the mucous membranes, *p* = 0.565. Polymorphous rash, *p* = 0.052. Changes in extremities, peripheral edema, *p* = 0.133. Cervical adenopathy, *p* = 0.04. ****** All patients (*n* = 301, 100%) had fevers for at least 5 days. *** Group 1: ≤1 year (*n* = 23), Group 2: ˃1 to ≤4 years (*n* = 177), and Group 3: >4 to ≤8 years (*n* = 101).

**Table 1 medicina-56-00145-t001:** Demographic data, disease timing, and KD clinical presentation in 301 children according to age at diagnosis.

Demographic Data	Group 1 (*n* = 23)	Group 2 (*n* = 177)	Group 3 (*n* = 101)
Sex (Male)	18 (78.3%)	101 (57.1%)	64 (63.4%)
Days between KD onset and diagnosis (minimum and maximum)	5–19 days	5–25 days	5–30 days
Days between KD onset and IVIG administration (minimum and maximum)	6–20 days	5–26 days	5–32 days
Incomplete or atypical KD presentation	9 (39.1%)	8 (4.5%)	7 (6.9%)

Group 1: ≤1 year, Group 2: ˃1 to ≤4 years, and Group 3: >4 years. KD: Kawasaki disease; IVIG: Intravenous immunoglobulin.

**Table 2 medicina-56-00145-t002:** Abnormalities during the convalescent and late phase of KD and other clinical findings.

Abnormalities	Group 1 (*n* = 23)	Group 2 (*n* = 177)	Group 3 (*n* = 101)	Total (*n* = 301)	*p* **
Behavioral abnormalities	2 (8.7%)	89 (50.3%)	73 (72.3%)	164 (54.5%)	0.000
Sensorineural hearing loss	0 (%)	29 (16.3%)	11 (10.8%)	41 (13.6%)	0.060
Beau’s lines	2 (8.7%)	57 (32.2%)	43 (42.6%)	102 (33.9%)	0.007
Nervous system abnormalities *	1 (4.3%)	3 (1.7%)	5 (5.0%)	9 (3.0%)	0.286
Disease recurrences	0 (0%)	21 (11.9%)	4 (4.0%)	25 (8.3%)	0.024
Coronary artery abnormalities	12 (52.2%)	38 (21.5%)	34 (33.7%)	84 (27.9%)	0.002
Abnormalities noncoronary arteries	2 (8.7%)	3 (8.7%)	3 (3.0%)	8 (2.7%)	0.157
Unresponsiveness to IVIG	6 (26.1%)	23 (13.0%)	14 (13.9%)	43 (14.3%)	0.240
Aseptic meningitis	2 (8.7%)	1 (0.6%)	0 (0%)	3 (1.0%)	0.019
Perianal desquamation	10 (43.5%)	102 (57.6%)	35 (34.7%)	147 (48.8%)	0.001
Pneumonitis	3 (13.0%)	54 (30.5%)	13 (12.9%)	70 (23.3%)	0.002
Gallbladder hydrops	0 (0%)	6 (3.4%)	0 (0%)	6 (2.0%)	0.156
Extreme Irritability	14 (60.9%)	129 (72.9%)	90 (89.1%)	233 (77.4%)	0.001
Diarrhea	5 (21.7%)	65 (36.7%)	26 (25.7%)	96 (31.9%)	0.093
Vomiting	6 (26.1%)	54 (30.5%)	27 (26.7%)	87 (28.9%)	0.763
Arthralgia, arthritis	0 (0%)	15 (8.5%)	15 (14.9%)	30 (10.0%)	0.062
Orchitis	0 (0%)	5 (2.8%)	5 (5.0%)	10 (3.3%)	0.467
Jaundice	0 (0%)	5 (2.8%)	3 (3.0%)	8 (2.7%)	0.804
Peripheral gangrene	1 (4.3%)	0 (0%)	1 (1.0%)	2 (0.7%)	0.066
Erythema and induration at BCG inoculation site	8 (34.8%)	5 (2.8%)	0 (0%)	13 (4.3%)	0.000

Group 1: ≤1 year, Group 2: >1 to ≤4 years, and Group 3: >4 years. * Seizures, chorea, facial paralysis, and learning deficit. ** Pearson chi-square test with Monte Carlo approximation. *** Red: the age groups did not differ statistically (*p* > 0.005). Blue: abnormalities or clinical findings with higher frequency in older age groups. Green: abnormalities or clinical findings with higher frequency in younger age groups. Black: the highest frequency appeared in the intermediate age group.

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
