# Peer review of "Clinical Manifestations of Kawasaki Disease at Different Age Spectrum: A Ten-Year Study"

_medicina, 2020, doi:10.3390/medicina56040145_

Round 1

Reviewer 1 Report

Manuscript ID: medicina-737881

Title: Clinical Manifestations of Kawasaki Disease at Different Age Spectrum: A Ten-year longitudinal study  

Minor comments and questions for the authors:  

  • On average how long did the behavior abnormalities persist at the studied population?
  • Did any of the children receive any other therapy than the one described in the study, like aspirin? 
  • What was the mean age of the children included in the study?
  • Were there any differences between gender of the patients?

Author Response

Review of manuscript ID medicina-737881 with title “Clinical Manifestations of Kawasaki Disease at Different Age Spectrum: A Ten-year longitudinal study”

Dear Editor,

Please see below the responses to reviewers' comments. In the revised manuscript, all the changes are highlighted using the "Track Changes" function in Microsoft Word. We hope that our paper is now suitable for publication. Thank you for the opportunity!

Reviewer #1:

Minor comments and questions for the authors:  

  • On average how long did the behavior abnormalities persist at the studied population?

R: We inserted the information on the manuscript after the table 2. The behavior abnormalities persist among six and twelve months.

  • Did any of the children receive any other therapy than the one described in the study, like aspirin? 

R: This information was previously described in the last paragraph of the results section. “Of the 301 patients in the study, 31 (10.2%) did not receive IVIG (2g / kg). Of the 270 who received IVIG, 31 patients (11.4%) did not respond to the first infusion and only responded to treatment with the second IVIG infusion. Nine children did not respond to the two pulses of IVIG (2g / Kg) and only became afebrile after pulses of methylprednisolone (30mg / Kg) for three days, and three patients also did not respond to the pulses of methylprednisolone and only became asymptomatic after using two doses of biological anti-TNF alpha (etanercept 0.8 mg / kg / week subcutaneous).”

  • What was the mean age of the children included in the study?

R: We inserted the information in the first paragraph of the results section. The mean age was 2.9 years.

  • Were there any differences between gender of the patients?

R: Despite the information about the number of males and females is describes in section 2.1. We inserted the information in the first paragraph of the results section to clarify the information. There was a proportion of 1.55 of males for each female.

Reviewer 2 Report

This article is an observational study of patients with Kawasaki disease for ten years in Latin America. This paper confirms the importance of early diagnosis of Kawasaki disease and enlightens the doctors who are not familiar with Kawasaki disease in Latin America.

Major

Although the observation period is ten years, the analytical technique is cross sectional.

The authors clarified the eligible criteria. However, the number of eligible patient, the number of patient studied and the number of patient dropped out during following up period are unclear.

The authors report important neurological complications of Kawasaki disease. Please discuss copmparison with previous reports from other ethnic groups. The authors mention only about recurrence ratio.

In this study, a marked delay in diagnosis exists. Could you show the incidence of the CALs at diagnosis of Kawasaki disease if you performed ultrasoud study before IVIG. Is there a relation between the day at diagnosis and the incidence of CALs at diagosis?

Page 7, Line 245- I guess most readers including me may not be familiar with the geography of Brasilia. Please describe the population that your institute covers at least.

Minor

There is confusion between digit separator(,) and decimal point(.).

Page 3, Line 97 CRP> 3,0 mg/dL,

Page 3, Line 100 albumin<or=3,0 g/dL

Page 3, Line 100 WBC count of>15.000/mm³ Also "of" is unnecesary.

Page 5, Line 167, Table 2, Line 6

   Sensorineural hearing loss 0 (%) 29 (16,3%) 11 (10.8%) 40 (13,6%) 0.060

Page 5, Line 179 in 41(13,6%) children

There is some mistake in English spelling.

Page 1, Line35 If you mention about Asian prevalence, please comment on the prevalence in Asian immigrants in your country.

Page 3, Line 98 I guess "ou" is Portuguese.

Page 3, Line 100  "/mm³" is missing in "or fever >or=450,000"

Page 5, Line 189 "IVIG(2g/Kg) " should be " IVIG (2g / kg)" like page6, line 192.

Page 6, Line 228 "disclose vd' is typing mistake.

Page 7, Line 274 "issu" is typing mistake.

Page 8, Line 318 The format of year/month is defferent in " Revista da Associacao Medica Brasileira (1992). 57(3):295–300".

Page 8, Line 332 The year/month is mission in "brasileira de reumatologia. 50(5):529–38. ". It should be "2010".

Author Response

Review of manuscript ID medicina-737881 with title “Clinical Manifestations of Kawasaki Disease at Different Age Spectrum: A Ten-year longitudinal study”

Dear Editor,

Please see below the responses to reviewers' comments. In the revised manuscript, all the changes are highlighted using the "Track Changes" function in Microsoft Word. We hope that our paper is now suitable for publication. Thank you for the opportunity!

Reviewer #2:

This article is an observational study of patients with Kawasaki disease for ten years in Latin America. This paper confirms the importance of early diagnosis of Kawasaki disease and enlightens the doctors who are not familiar with Kawasaki disease in Latin America.

Major

Although the observation period is ten years, the analytical technique is cross sectional.

 R: Ok. We changed it in the manuscript.

The authors clarified the eligible criteria. However, the number of eligible patient, the number of patient studied and the number of patient dropped out during following up period are unclear.

R: We inserted the information in the manuscript (section 2.1). The number of patients studied was 301, those who were liost to follow-up were few in the first yar of the study were not included.

The authors report important neurological complications of Kawasaki disease. Please discuss copmparison with previous reports from other ethnic groups. The authors mention only about recurrence ratio.

R: We inserted the information in the manuscript discussion section. Few studies have evaluated these changes. Most studies on neurological disorders are case reports or case studies[1–3]. These studies describe neurological changes based on an area of cerebral ischemia or involvement of the facial nerve with facial paralysis, cochlear nerve leading to ataxia, and impairment of the acoustic nerve with sensorineural hearing loss[1–3]. Of the 301 patients, only 24 (8%) were of Asian descent (first and second degree). A study based on ethnic groups was not performed since, in Brazil, there is large miscegenation of ethnic groups, making difficult the classification. KD recurrence occurred in 25 (8.3%) of children during the first three years of follow-up. In seven patients, KD recurred twice with an interval between episodes ranging from 4 months to two years.

Reference:

  1. McCrindle, B. W.; Rowley, A. H.; Newburger, J. W.; Burns, J. C.; Bolger, A. F.; Gewitz, M.; Baker, A. L.; Jackson, M. A.; Takahashi, M.; Shah, P. B.; Kobayashi, T.; Wu, M. H.; Saji, T. T.; Pahl, E. Diagnosis, treatment, and long-term management of Kawasaki disease: A scientific statement for health professionals from the American Heart Association. Circulation 2017, 135, e927–e999, doi:10.1161/CIR.0000000000000484.
  2. Smith, K. A.; Yunker, W. K. Kawasaki disease is associated with sensorineural hearing loss: A systematic review. Int. J. Pediatr. Otorhinolaryngol. 2014, 78, 1216–1220, doi:10.1016/j.ijporl.2014.05.026.
  3. Magalhães, C. M. R.; Magalhães Alves, N. R.; Oliveira, K. M. A.; Silva, I. M. C.; Gandolfi, L.; Pratesi, R. Sensorineural hearing loss: An underdiagnosed complication of Kawasaki disease. J. Clin. Rheumatol. 2010, 16, 322–325, doi:10.1097/RHU.0b013e3181f603bc.

In this study, a marked delay in diagnosis exists. Could you show the incidence of the CALs at diagnosis of Kawasaki disease if you performed ultrasoud study before IVIG. Is there a relation between the day at diagnosis and the incidence of CALs at diagosis?

 R: We inserted the information in paragraph before table 2. Among the 84 children that presented with CALs, the abnormalities were observed during the acute phase in 18 (21.4 %), during the subacute phase in 49 (58.3 %), and during the late phase of the disease in 17 (20.3 %). In the 18 patients who had coronary changes in the acute phase, the echocardiogram was performed a few hours before (which motivated the use of IGEV) or a few hours later, because after the use of IGEV the acute phase ends with the disappearance of fever and others symptoms like, also normalize the inflammatory tests.

Page 7, Line 245- I guess most readers including me may not be familiar with the geography of Brasilia. Please describe the population that your institute covers at least.

R: We inserted the information as a footnote. “Brasilia is the capital of Brazil, located in the Federal District / Midwest Brazilian Region. Around Brasilia, there are cities with disordered growth and some areas with low-income people and poor housing conditions. The research was carried out at Child Hospital, which is a public hospital located in Brasilia that is part of the Unified Health System and serves low and middle-income patients in Brasilia and the surrounding area.”

Minor

There is confusion between digit separator(,) and decimal point(.).

Page 3, Line 97 CRP> 3,0 mg/dL,

Page 3, Line 100 albumin< or =3,0 g/dL

Page 3, Line 100 WBC count of >15.000/mm³ Also "of" is unnecesary.éPage 5, Line 167, Table 2, Line 6

   Sensorineural hearing loss 0 (%) 29 (16,3%) 11 (10.8%) 40 (13,6%) 0.060

Page 5, Line 179 in 41(13,6%) children

R:  We are sorry for our mistake. We corrected it in the manuscript.

There is some mistake in English spelling.

Page 1, Line35 If you mention about Asian prevalence, please comment on the prevalence in Asian immigrants in your country.

R: The prevalence of Asian immigrants in Brazil is higher in the southeastern and southern regions. In the Federal District and its surroundings (Midwest Brazilian Region), where the study was conducted, the number of Asian immigrants is still small. Of the 301 patients, only 24 (8%) were of Asian descent (first and second degree). As we mentioned before, a study based on ethnic groups was not performed since, in Brazil, there is large miscegenation of ethnic groups, making difficult the classification. We inserted the information in the 4th paragraph of the discussion section.

Page 3, Line 98 I guess "ou" is Portuguese.

R: We corrected it.

Page 3, Line 100  "/mm³" is missing in "or fever >or=450,000"

R: We corrected it.

Page 5, Line 189 "IVIG(2g/Kg) " should be " IVIG (2g / kg)" like page6, line 192.

R: We corrected it.

Page 6, Line 228 "disclose vd' is typing mistake.

R: We corrected it.

Page 7, Line 274 "issu" is typing mistake.

R: We corrected it.

Page 8, Line 318 The format of year/month is defferent in " Revista da Associacao Medica Brasileira (1992). 57(3):295–300".

R: We corrected it.

Page 8, Line 332 The year/month is mission in "brasileira de reumatologia. 50(5):529–38. ". It should be "2010".

R: We corrected it.